# Autophagy Promotes Infectious Particle Production of Mopeia and Lassa Viruses

**DOI:** 10.3390/v11030293

**Published:** 2019-03-23

**Authors:** Nicolas Baillet, Sophie Krieger, Alexandra Journeaux, Valérie Caro, Frédéric Tangy, Pierre-Olivier Vidalain, Sylvain Baize

**Affiliations:** 1Unité de Biologie des Infections Virales Emergentes, Institut Pasteur, 69007 Lyon, France; nicolas.baillet@hotmail.fr (N.B.); sophieeve@live.fr (S.K.); alexandra.journeaux@pasteur.fr (A.J.); 2Centre International de Recherche en Infectiologie (INSERM U1111, CNRS UMR5308, ENS Lyon, Université Lyon I), 69007 Lyon, France; 3Unité Environnement et Risques Infectieux, CIBU, Infection et Epidémiologie, Institut Pasteur, 75015 Paris, France; valerie.caro@pasteur.fr; 4Unité de Génomique Virale et Vaccination, Institut Pasteur, CNRS UMR-3569, 75015 Paris, France; frederic.tangy@pasteur.fr (F.T.); pierre-olivier.vidalain@parisdescartes.fr (P.-O.V.)

**Keywords:** LASV, MOPV, Z matrix protein, NDP52, TAX1BP1, autophagy, replication, yeast two-hybrid screening

## Abstract

Lassa virus (LASV) and Mopeia virus (MOPV) are two closely related Old-World mammarenaviruses. LASV causes severe hemorrhagic fever with high mortality in humans, whereas no case of MOPV infection has been reported. Comparing MOPV and LASV is a powerful strategy to unravel pathogenic mechanisms that occur during the course of pathogenic arenavirus infection. We used a yeast two-hybrid approach to identify cell partners of MOPV and LASV Z matrix protein in which two autophagy adaptors were identified, NDP52 and TAX1BP1. Autophagy has emerged as an important cellular defense mechanism against viral infections but its role during arenavirus infection has not been shown. Here, we demonstrate that autophagy is transiently induced by MOPV, but not LASV, in infected cells two days after infection. Impairment of the early steps of autophagy significantly decreased the production of MOPV and LASV infectious particles, whereas a blockade of the degradative steps impaired only MOPV infectious particle production. Our study provides insights into the role played by autophagy during MOPV and LASV infection and suggests that this process could partially explain their different pathogenicity.

## 1. Introduction

Lassa virus (LASV), a mammalian Old-World arenavirus, is the etiological agent of Lassa fever (LF), an acute and occasionally severe viral hemorrhagic fever in humans. The disease is a significant cause of morbidity (100,000–300,000 infections annually) and mortality (about 5000 fatalities annually) and is constrained to sub-Saharan West Africa [1]. Humans are infected after cutaneous-mucosal contact with contaminated fluids or feces from the natural host of LASV, the peridomestic rodent *Mastomys sp.* [2]. Direct exposure to body fluids or contaminated materials from infected patients is responsible for human-to-human transmission, especially during nosocomial outbreaks, but most of the infections that occur during outbreaks result from reservoir-to-human transmission [3,4]. The absence of a licensed vaccine and efficient antiviral drug available in endemic countries renders LF a public health problem, exacerbated by the current expansion of the zone of endemicity [5]. Mopeia virus (MOPV) also belongs to the Old-World complex of mammarenaviruses and is very closely related to LASV in addition to being hosted by natal rat. However, no human case of MOPV infection has ever been reported [6]. MOPV has even been shown to confer protection against a challenge with LASV in non-human primates and therefore represents a useful platform to generate protective vaccines against Lassa fever [7,8,9]. Comparing LASV and MOPV should therefore allow the identification of immune and viral features involved in LF pathogenesis.

The bi-segmented RNA genome of the Arenaviridae family encodes four proteins: the nucleoprotein NP, the surface glycoprotein GP, the polymerase L, and the RING finger protein Z [10,11,12]. Although it is the smallest arenavirus protein, the Z-matrix protein has multiple functions in the viral life cycle, including viral assembly [13,14,15,16,17,18] and budding [19,20,21], transcriptional repression [22,23,24], interferon antagonism [17,25,26] and interactions with multiple host-cell proteins, notably proteins involved in the ESCRT machinery [19,20,27], promyelocytic leukemia protein (PML) [28], ribosomal protein P0 [28], eukaryotic translation initiation factor 4E (eIF4E) [29] and proline-rich homeodomain protein (PRH) [30]. Overall, the arenavirus Z protein is central in the viral life cycle and interacts with a variety of cellular factors that have been partially discovered, but are not yet fully understood. These associations may trigger or hijack numerous cell pathways that facilitate viral replication and could explain the differences in pathogenicity among members of the Arenaviridae family.

Among the cell pathways counteracted by pathogens is autophagy. Autophagy is an evolutionarily conserved catabolic process essential for the maintenance of cellular homeostasis through the elimination of senescent cytosolic components and the recycling of metabolites [31]. The process involves a lysosomal-dependent mechanism initiated by the formation of an isolated membrane, the phagophore, which elongates around cytosolic components to generate a newly-formed vesicle called the autophagosome. The fusion of autophagosomes with lysosomes forms mature autolysosomes in which degradation occurs. The initiation, elongation, and regulation steps of the autophagy process rely on autophagy-related (ATG) proteins. The autophagy machinery is also a part of the host defense system as it contributes to the degradation of invading pathogens by delivering them to the lysosomal compartment [32]. Autophagy may also deliver intracellular pathogen-associated molecular patterns (PAMPs) to endosomal-pattern recognition receptors (PRRs) and MHC-loading compartments, therefore contributing to activate innate [33,34] and adaptive [35,36,37] antiviral immune responses. As obligate intracellular pathogens, viruses have evolved various strategies to escape, inhibit, or hijack the autophagic pathway to evade immune responses and favor viral replication [38]. However, the role of autophagy during MOPV and LASV, and more generally during arenavirus infection, is still unknown. 

Here, we performed a yeast two-hybrid (Y2H) screening to identify cellular partners of the Z protein of the pathogenic LASV and non-pathogenic MOPV. We identified two autophagy receptors, calcium-binding and coiled-coil domain 2 (CALCOCO 2 or NDP52) and TAX1BP1 (or CALCOCO 3), suggesting a link between autophagy and viral infection. We then monitored autophagy flux in infected cells and observed an increase of this flux in MOPV- but not LASV-infected cells. We show that early steps of autophagy are important in the late stages of replication for LASV and MOPV, whereas the degradation steps of autophagy benefits only MOPV infectious particle production. These data provide insight on the proviral role played by autophagy during LASV and MOPV infection and we discussed how it may contribute to their different pathogenicity.

## 2. Materials and Methods

### 2.1. Yeast Two-Hybrid Screening

Yeast two-hybrid screens were performed following the protocol described in Vidalain et al. [39]. DNA sequences encoding the matrix proteins (Z) of LASV or MOPV were cloned by in vitro recombination (Gateway technology; Invitrogen, Carlsbad, CA, USA) from pDONR207 into the yeast two-hybrid vector pPC97-GW for expression in fusion downstream of the GAL4 DNA-binding domain (GAL4-BD). AH109 yeast cells (Clontech; Takara, Mountain View, CA, USA) were transformed with these baits using a standard lithium-acetate protocol. Spontaneous transactivation of the HIS3 reporter gene was observed in yeast cells expressing GAL4-BD-Z. Consequently, screens were performed on a synthetic medium lacking histidine (-His) and supplemented with 3-amino-1,2,4-triazole (3-AT) at 5 to 10 mM for MOPV and 80 to 100 mM for LASV. A mating strategy was used to screen three different prey libraries with distinct characteristics: a human spleen cDNA library, a mouse brain cDNA library, and a normalized library containing 12,000 human ORFs. All libraries were established in the yeast two-hybrid expression plasmid pPC86 to express prey proteins in fusion downstream of the GAL4 transactivation domain (GAL4-AD). After six days of culture, colonies were picked, replica plated, and incubated over three weeks on selective medium to eliminate potential contamination with false positives. Prey proteins from selected yeast colonies were identified by PCR amplification using primers that hybridize within the pPC86 regions flanking the cDNA inserts. PCR products were sequenced, and cellular interactors were identified by multi-parallel BLAST analysis.

### 2.2. Gap-Repair Procedure

PCR products were co-transformed into AH109 yeast cells expressing GAL4-BD-Z constructs together with an empty pPC86 vector linearized downstream of the GAL4-AD coding sequence [40]. Homologous recombination in yeast cells between PCR products and linearized pPC86 vectors allows the reconstruction of the GAL4-AD-Prey sequences. Transformed cells were plated on selective -His media supplemented with 3AT at 5 mM for the MOPV-Z protein, and 100 mM for the LASV-Z protein. After five days of culture on selective medium, growing colonies were scored.

### 2.3. Cell Lines and Viruses

VeroE6 cells were maintained in DMEM supplemented with 0.5% penicillin-streptomycin (PS) and 5% fetal bovine serum (FBS). The HeLa and 293T cell lines were maintained in DMEM supplemented with 0.5% PS and 10% FBS. The GFP-LC3 HeLa cell line (kindly provided by Dr Mathias Faure, Centre International de Recherche en Infectiologie (CIRI), Lyon, France) was grown in RPMI supplemented with 0.5% PS and 10% FBS (all from Invitrogen, Carlsbad, CA, USA). Mopeia (AN21366 strain [6]) and Lassa (AV strain [41]) viruses were cultivated in VeroE6 cells at 37°C in 5% CO_2_. Viral supernatants were harvested and used as the virus stock and the absence of mycoplasma was confirmed. LASV and MOPV titers were determined by plaque immunoassays as described below. All experiments with LASV were carried out in biosafety level 4 facilities (Laboratoire P4 Jean Mérieux-Inserm, Lyon). 

### 2.4. Plasmids, Antibodies, and Reagents

ZM-FLAG and ZL-FLAG were cloned into a pHCMV vector, allowing the expression of MOPV or LASV Z matrix protein with a FLAG tag in the C-terminal position (inserted between the XmaL and BamHI restriction sites). The same constructions were made for ZM and ZL-mCherry plasmids. TAX1BP1 plasmids were engineered into pEGFP-C1, allowing the expression of GFP-tagged proteins for immunofluorescence experiments. eGFP-NDP52 was a kind gift from Dr Mathias Faure (CIRI, Lyon, France). The primary antibodies used were: anti-actin (A3854), anti-FLAG (A8592), anti-ATG5 (A0856), and anti-TAX1BP1 (HPA024432), all from Sigma-Aldrich (Saint Quentin Fallavier, France); anti-NDP52 (ab68588) from Abcam (Cambridge, MA, USA); anti-GAPDH (sc-25778, Santa Cruz Biotechnology, Dallas, TX, USA), anti-p62 (610832, BD Biosciences, San Jose, CA, USA), and anti-Z (Agro-Bio, La Ferté Saint-Aubin, France). The secondary antibodies used for Western blotting were anti-mouse conjugated to peroxidase (111-035-174) and anti-rabbit conjugated to peroxidase (111-035-144), all from Jackson ImmunoResearch (Cambridge House, UK). The secondary antibody used for confocal microscopy was an anti-mouse conjugated to Alexa 555 (A21424) from Invitrogen (Carlsbad, CA, USA). The pharmacological agent used was chloroquine (C6628), purchased from Sigma-Aldrich.

### 2.5. RNAi Analysis

Freshly passaged HeLa cells were added to pre-plated transfection media containing the indicated siRNA at a final concentration of 20 nM and using Lipofectamine RNAiMAX (Invitrogen, Carlsbad, CA, USA), according to the manufacturer’s instructions. The cells were maintained for 72 h after siRNA transfection in six-well plates at 1.2 × 10^5^ cells per well. The cells were then counted and infected with MOPV or LASV (MOI = 0.1) for 1 h in DMEM medium supplemented with 2% FBS and 0.5% PS. Infectious medium was removed before adding fresh medium (DMEM 5% FBS, 0.5% PS) for the indicated times of the experiment. Viral RNA and infectious particles were then quantified (please see corresponding paragraph). Silencing RNAs against NDP52 (siNDP52) and TAX1BP1 (siTAX1BP1) were purchased from Ambion (Thermo Fisher Scientific, Waltham, MA, USA) (references 4392421 and 4392420, respectively). Silencing RNAs against ATG5 (M-004374-04) were purchased from Dharmacon (Lafayette, CO, USA), as well as the non-targeting control siRNA (D-001210-5-5). The efficiency of each siRNA was validated by Western blotting.

### 2.6. Coimmunoprecipitation and Western Blot Analysis

Human 293T cells (ATCC, LGC Standards, Molsheim, France) were transfected with the indicated plasmids using Lipofectamine 2000 (Invitrogen), according to the manufacturer’s protocol. At 15 h post-transfection, cells were harvested and lysed in non-denaturing lysis buffer (10 mM Hepes, 5 mM EDTA, 150 mM NaCl, 1% NP40, and 50 µM PR-619 (662141, Calbiochem; Merck, Darmstadt, Germany) supplemented with protease inhibitors (11873580001, Sigma-Aldrich). Lysates were clarified for 10 min at 10,000 rpm. Supernatants were then incubated with M2 anti-FLAG magnetic beads (M8823, Sigma-Aldrich) for 2 to 3 h at 4 °C under agitation. The beads were washed four times in lysis buffer before boiling 15 min in loading buffer. For Western blots, equal amounts of protein were loaded and separated on 4–15% gradient precast gels and transferred onto PVDF membranes before staining. Samples were immunoblotted with the appropriate primary antibodies. Protein levels were quantified by measuring the intensity of the bands by densitometry. Actin or GAPDH were used as positive controls of the cell extracts. 

### 2.7. Microscopy Analysis

All images were acquired using a confocal LSM 510 (Zeiss, Oberkochen, Germany) with an Axioscope 63× oil immersion lens objective. The images were analyzed using ImageJ/Fiji (version 1.52m). HeLa or GFP-LC3 HeLa cells were cultured in 12-well plates with a sterile coverslip in each well. The cells were fixed in 4% paraformaldehyde for 20 min and treated with glycine (0.1 M). For Z protein-autophagosome colocalization experiments, we performed additional steps comprised of lysis (0.1% Triton X-100), saturation (PBS, 2% Bovine Albumin Serum (BSA), 5% glycerol, and 0.1% Tween 20), and primary antibody staining (mouse anti-Flag, Sigma-Aldrich), followed by staining with a secondary antibody conjugated to Alexa Fluor 555. All samples were mounted with DAPI (P36931, Invitrogen). For the counting of autophagosomes, GFP-LC3^+^ vesicles were enumerated for 50 cells for each condition. 

### 2.8. Viral Entry Assays

Twelve-well plates were seeded with 1.5 × 10^5^ HeLa cells per well and the cells treated the day after with 50 µM chloroquine (CQ) for 2 h before being placed on ice for 30 min. Cells were then infected with MOPV at an MOI of 2 and placed on ice for 30 min before transfer to 37 °C (heat shock). Cells were maintained for the indicated times before 0.25% trypsin treatment (Gibco; Thermo Fisher Scientific, Waltham, MA, USA) for 30 min, in order to remove a non-internalized virus. Cells were then centrifuged for 5 min at 1300 rpm and washed two times with PBS. After centrifugation, pellets were lysed using the RNeasy Minikit (Qiagen, Hilden, Germany) for quantitative viral analysis. 

### 2.9. Quantitative Viral RNA Analysis

Viral RNA was extracted from culture supernatants and cells using the QIAamp Viral RNA and RNeasy Minikits, respectively (all from Qiagen), according to the manufacturer’s instructions. Quantitative PCR for viral RNA was performed with the EuroBioGreen Lo-ROX qPCR mix (Eurobio, Les Ulis, France), using primers 5’-CTTTCCCCTGGCGTGTCA-3’ and 5’-GAATTTTGAAGGCTGCCTTGA-3’ for MOPV and 5’-CTCTCACCCGGAGTATCT-3’ and 5’-CCTCAATCAATGGATGGC-3’ for LASV.

### 2.10. Viral Infection and Titration

Vero cells were infected with sequential dilutions of supernatant and incubated at 37 °C in 5% CO_2_ for six days with Carboxy-methyl-cellulose (1.6%) (BDH Laboratory Supplies, Poole, UK) in DMEM supplemented with 2% FBS. Infectious foci were detected by incubation with monoclonal antibodies (mAbs) directed against MOPV and LASV (mAbs L52-54-6A, L53-237-5 and YQB06-AE05, generously provided by Dr P. Jahrling, (USAMRIID, Fort Detrick, MD USA), followed by PA-conjugated goat polyclonal anti-mouse IgG (Sigma-Aldrich).

### 2.11. Statistical Analysis

Statistical analyses were performed using GraphPad Prism (version 8.0.2, GraphPad Software, San Diego, CA, USA). Data were analyzed using Student’s *t*-test or the non-parametric Mann–Whitney test to determine the significance of differences (**p* < 0.05, ***p* < 0.01, ****p* < 0.001, n.s. non-significant).

## 3. Results

### 3.1. Yeast two-hybrid (Y2H) Screening of LASV and MOPV Z Matrix Proteins Reveal Host Cell Interactors Involved in Autophagy Pathways

We identified host cellular proteins that interact with Z proteins of both viruses by using a high throughput yeast two-hybrid (HT-Y2H) screening protocol in which AH109 yeast cells were transformed with a Y2H vector encoding the Z protein of LASV or MOPV fused to the GAL4-BD domain (viral bait). Y187 yeast cells were transformed with prey libraries consisting of a human spleen cDNA library, mouse brain cDNA library, or normalized library containing 12,000 human ORFs fused to the GAL4-AD domain (cDNA library prey). Interactions between prey and bait lead to transactivation of the *HIS3* reporter gene. After mating AH109 and Y187 yeast on selective medium lacking histidine, the prey proteins from positive colonies were identified by PCR amplification. The PCR products were sequenced, and the cellular interactors identified by multi-parallel BLAST analysis. Y2H screens allowed us to identify, among others, two known autophagy adaptors, namely NDP52 and TAX1BP1, which have been shown to interact with the Z protein of MOPV, but not LASV (Table 1). Interactions between TSG101 and arenavirus Z protein have already been reported by others and were used here to validate our screen [19,42]. NDP52 and TAX1BP1 are autophagy adaptors, known as Sequestosome 1-like receptors (SLRs). SLRs recognize motifs (such as ubiquitin or galectin) present on the surface of invading pathogens through a ubiquitin-binding domain (UBD) and target them into autophagosomes through a LC3-interacting region (LIR) for selective degradation by autophagy [43]. We then confirmed the results by retesting each interaction in the Y2H system using the gap-repair procedure, in which DNA sequences encoding both viral Z proteins and candidate host interactors are introduced into fresh yeast cells. Transformed cells were plated on selective medium lacking histidine and the growing colonies scored after five days of culture on selective medium (Figure 1A). Analysis of the colonies confirmed the previous results obtained by Y2H. These findings suggest that NDP52 and TAX1BP1 may play an important role in the life cycle of MOPV or LASV. Moreover, they also suggest that autophagy may have different roles during LASV and MOPV infection as NDP52 and TAX1BP1 interacted only with the MOPV Z protein. 

### 3.2. NDP52 and TAX1BP1 Interact with the Z Proteins of LASV and MOPV

We next wanted to confirm the novel interactions identified by Y2H in mammalian cells. 293T cells were transfected with LASV or MOPV C-terminal tagged FLAG Z protein and/or an N-terminal eGFP-fused NDP52 (eGFP-NDP52) or TAX1BP1 (eGFP-TAX1BP1) protein (Figure 1B). Cell lysates were directly analyzed by Western blotting (input) or immunoprecipitated (IP) with anti-FLAG coupled magnetic beads followed by Western blot analysis of coimmunoprecipitated eGFP-tagged proteins. Surprisingly, NDP52 and TAX1BP1 coimmunoprecipitated with the Z protein of both LASV and MOPV, contrary to the results of the Y2H screen. This discrepancy may be due to the 3-AT concentration used, which was 10 to 20 times higher for the LASV Z screens in the selective media of the Y2H and gap-repair screening. We then assessed the colocalization of Z and the previously described interactors in another mammalian cell line. Briefly, HeLa cells were transfected with the indicated plasmids before being fixed and analyzed by confocal microscopy. The Z proteins of MOPV and LASV colocalized with eGFP-TAX1BP1 (Figure 1C) and eGFP-NDP52 (Figure 1D) in the cytoplasm of transfected Hela cells. Overall, our findings show that both LASV and MOPV Z matrix proteins interact with NDP52 and TAX1BP1 in transfected 293T and HeLa cells. 

### 3.3. Reduced Expression of NDP52 or TAX1BP1 Has No Effect on LASV and MOPV Replication in HeLa Cells

We next evaluated the functional impact of NDP52 and TAX1BP1 on the LASV and MOPV life cycles. Briefly, HeLa cells were transfected with control non-targeting siRNA, siNDP52, or siTAX1BP1. TAX1BP1 and NDP52 protein levels were much lower in siTAX1BP1 and siNDP52 transfected cells, respectively, than in cells transfected with non-targeting siRNAs 72 h after transfection (Figure 2A). The silenced cells were then infected with LASV or MOPV at an MOI of 0.1. After three days in culture, we evaluated the quantity of extracellular and intracellular RNA, as well as that of viral infectious particles. There were no significant differences in the amount of intracellular viral RNA (Figure 2B) or infectious particles (Figure 2C) between siCTL, siNDP52, or siTAX1BP1 transfected cells for either LASV or MOPV. However, there was slightly less extracellular LASV viral RNA for siTAX1BP1 transfected cells than those transfected with siCTL, whereas the level of MOPV viral RNA was unaffected (Figure 2D). Overall, these results suggest that NDP52 and TAX1BP1 are not involved in the production of infectious virus for either MOPV or LASV in HeLa cells. 

### 3.4. Autophagy Is Induced during MOPV but Not LASV Infection in HeLa Cells

We thus sought to determine whether autophagy was involved in the infection of MOPV or LASV in HeLa cells. As NDP52 and TAX1BP1 interact with Z and are known to be involved in the selective degradation of pathogens by autophagy, we first assessed whether MOPV or LASV Z protein is present in autophagosomes by transfecting MOPV or LASV Z protein into HeLa cells that stably express a GFP-fused LC3 (GFP-LC3 HeLa), commonly used as a marker of autophagosomes (Figure 3A) [46]. GFP-LC3 HeLa cells were treated with chloroquin (CQ) for two hours before transfection with Z plasmids in order to prevent the degradation of autophagosomes, thus increasing the probability to observe the possibly internalized matrix protein. Strikingly, both MOPV and LASV Z proteins colocalized with GFP-LC3 puncta, suggesting a link between the autophagy machinery and MOPV or LASV infection. We therefore examined the kinetics of autophagy upon infection with either virus. We infected HeLa cells with LASV or MOPV at an MOI of 2 and assessed autophagy at various timepoints by evaluating the level of SQSTM1/p62, a long-lived protein mainly degraded through autophagy; its degradation in cells indicating an effective autophagy flux (Figure 3B) [46]. There was a large decrease in p62 levels two days after MOPV infection, suggesting an increase of the autophagy flux at this late timepoint. In contrast, p62 levels did not change in the LASV infected cells. We then sought to confirm the induction of autophagy in MOPV-infected GFP-LC3 HeLa cells by counting the number of GFP-LC3-labeled puncta, representing autophagosomes, over time (Figure 3C). Consistent with the previous findings, MOPV infection induced a transient accumulation of autophagosomes two days after infection in GFP-LC3 HeLa cells. Overall, these data strongly suggest that MOPV, but not LASV, induces a transient wave of autophagy two days after infection in HeLa cells. 

### 3.5. Early Stages of Autophagy Promote MOPV and LASV Infectious Particle Production

Conversely to several viruses that take advantage of autophagy to replicate, while inhibiting autophagosome maturation, MOPV (but not LASV) infection induced an increase in the autophagy flux. We investigated whether the autophagy flux is involved in MOPV or LASV infectious particle production. First, we impaired the early steps of autophagy in MOPV- or LASV-infected cells by reducing the expression of ATG5, essential for autophagosome formation (Figure 4A). We confirmed that silencing of ATG5 was efficient 72 h after transfection, before infecting the cells with MOPV or LASV at an MOI of 0.1. RT-qPCR and the titration of infectious particles in cells and/or supernatants revealed significantly less viral RNA (Figure 4B) and fewer infectious viral particles (Figure 4C) in the supernatants of HeLa cells infected by MOPV, and surprisingly LASV, when the expression of ATG5 was impaired as compared to control conditions. However, the quantity of viral RNA inside the cells was not affected by the silencing of ATG5 for either virus (Figure 4D), suggesting that the autophagy flux did not affect the viral entry or replication steps. Altogether, these results indicate that the early stages of autophagy play an important role during the late stages of the MOPV and LASV life cycle. 

### 3.6. The Degradation Steps of Autophagy Affect Only MOPV Infectious Particle Production

We next sought to confirm the impact of autophagy on infectious viral particle production by treating cells with CQ. CQ prevents endosomal acidification and is a well-known inhibitor of autophagosome maturation, which impairs both the fusion of autophagosomes with lysosomes and lysosomal protein degradation. Viral entry of arenaviruses is characterized by internalization of the virus by endocytosis into acidified endosomes, where viral fusion occurs at low pH [47]. We performed a viral entry assay to ensure that we were affecting autophagy rather than viral entry when using CQ in the HeLa cells. We pretreated cells with 50 µM CQ and placed them on ice before infection with MOPV at a high MOI (2) (Figure 5A). We subjected the infected cells to heat shock and quantified the internalized viral RNA by RT-qPCR at various timepoints. There were no differences in the quantity of internalized MOPV RNA between CQ-pretreated and untreated cells, indicating that CQ did not affect viral internalization by HeLa cells. 

We investigated the role of the late stages of autophagy during MOPV or LASV infection by infecting HeLa cells with LASV or MOPV at an MOI of 0.1. One day after infection, cells were treated, or not, with CQ at a final concentration of 50 µM for 48 h. Three days after infection, cell supernatants and lysates were harvested. The inhibition of autophagosome maturation with CQ significantly decreased the infectious particle production of MOPV, but not that of LASV, when relative to control conditions (Figure 5B). In addition, the treatment of cells with CQ significantly decreased the level of Z protein expression in MOPV- but not LASV-infected cells (Figure 5C). Overall, these results indicate that MOPV infection leads to productive autophagy which is required for the efficient production of infectious particles. In contrast, LASV infection does not trigger autophagy but basal autophagy is still important to improve the production of LASV infectious particles.

## 4. Discussion

Arenavirus Z protein is a critical regulator that mediates the budding steps prior to virus release, ensuring the packaging of all viral components required for infectious particle production [19,20,27]. To this end, the Z protein recruits several host proteins to facilitate efficient release of viral progeny from the plasma membrane of the infected cell. We chose to screen the LASV and MOPV matrix-protein host interactors through Y2H screening because of its simplicity, its high throughput analysis, and the possibility to identify direct protein–protein interactions, in contrast to protein complex analysis by mass spectrometry, which does not discriminate direct from indirect partners. However, Y2H could represent a bias as Z protein is known to localize to cell membranes. It is thus possible that some interactions between Z and host cell proteins that take place in specific subcellular membranous compartments are unlikely to be recreated in the Y2H screen performed in this study. This method allowed us to isolate a number of host interactors and we selected from among them autophagy adaptors: NDP52 and TAX1BP1. Y2H and gap-repair data identified NDP52 and TAX1BP1 as exclusive MOPV Z protein interactors, but immunoprecipitation assays and confocal microscopy observations revealed that the LASV Z protein is also a partner of the two autophagy receptors. This difference in the detection threshold may be due to the 3-AT concentration used (10 to 20 times higher for the LASV Z screens) in the selective media for Y2H and gap-repair screening. 3-AT is a competitive inhibitor of the HIS3 enzyme and is used to negatively select yeast growth when the bait protein alone (Z) self-transactivates the *HIS3* gene, thus limiting false-positive interactions. In our case, the self-transactivation level of the LASV Z bait protein was higher than that of the MOPV Z bait protein. We thus used a more stringent selective medium for the LASV Z constructions, which may have limited the true positive interactions between bait and prey plasmids in the yeast. We finally identified NDP52 and TAX1BP1 as interactors of both the LASV and MOPV Z proteins, as the confirmation by immunoprecipitation assays was performed under the same conditions for the matrix proteins of both viruses. Other host factors interacting with LASV Z protein and possibly linked to autophagy have been unraveled by others [48]. In this study, authors have identified the WW-domain bearing protein BCL2 Associated Athanogene 3 (BAG3) as a negative regulator of Lassa VLP egress. BAG3 has several functions in cell cycle including a stress-induced role in chaperone-assisted selective autophagy (CASA). Although this study appears contradictory to the proviral role of autophagy described here during LASV infection, it reinforces the link existing between Mammarenavirus infection and autophagy pathways.

We then analyzed the functional role of NDP52 and TAX1BP1 in the viral life cycle. These two proteins are close paralogs involved in selective autophagy, in which specific cargo is first tagged by ubiquitination following their recognition by autophagy adaptor proteins for subsequent targeting to autophagosomes for degradation [49]. This phenomenon has been well studied in bacteria and in several studies for viruses. For example, both TAX1BP1 and NDP52 can affect the replication of Measles virus by interacting with viral proteins and facilitating the maturation of autophagosomes [50]. Another autophagic adaptor, p62, has been shown to mediate autophagic degradation of Chikungunya virus in murine cells [51]. Here, we show that NDP52 and TAX1BP1 are neither involved in the replication nor the release of MOPV or LASV infectious particles, despite their interactions with the Z proteins, as shown by RNAi interference assays. These results were unexpected and suggest that autophagy adaptors could play a role during LASV or MOPV infection that does not affect the viral fitness in the cell lines we used. One possible role of this interaction could be the regulation of inflammation, as TAX1BP1 and NDP52 have been shown to downregulate NFκB and IRF3 signaling pathways [52,53]. The nucleoprotein of LASV is known to interfere with type I interferon (IFN-I) production through the sequestration of the IKK complex, leading to inhibition of the IRF3 pathway [54]. Thus, both Z and NP could act in concert to inhibit the IKK complex, therefore controlling inflammatory response in infected cells. 

We extended our observations by investigating the role played by autophagy in transfected and infected cells. First, we show that the LASV and MOPV Z proteins colocalize with GFP-LC3 vesicles, arguing that autophagy likely plays a role during the course of the infection of both viruses. This observation, in conjunction with the large decrease in the quantity of Z protein in MOPV-infected cells treated with CQ, suggests that the Z protein is targeted towards autophagic degradation. Such sequestration of the matrix protein in autophagosomes could have a role in Z protein processing and viral assembly, as shown for Gag in HIV-infected macrophages [55]. However, sequestration of the Z protein has been observed in cells over-expressing it via plasmid transfection, which may not accurately recreate the situation found in virus-infected cells. We then demonstrated that MOPV, but apparently not LASV, induces a transient wave of autophagy two days after infection. These experiments were made using a high MOI (2) for each virus in order to monitor autophagy flux when most cells in the culture are infected at the same time. The correlation between p62 turnover and the higher number of GFP-LC3 vesicles indicates that MOPV-induced autophagy is effective. We could not investigate whether LASV induced accumulation of autophagosomes in GFP-LC3 HeLa cells because of the lack of confocal microscopy within BSL-4 laboratory. However, the absence of a difference in p62 levels over time in LASV-infected cells suggests that LASV does not affect autophagy flux as p62 is generally degraded in autolysosomes. How autophagy is induced during MOPV infection is unknown. As with others mammarenaviruses, MOPV establishes long-term persistent infections in their natural hosts. Under the experimental conditions we used in this study, MOPV may transiently trigger the UPR-ER stress pathway, which could then induce autophagy. This phenomenon has been highlighted during HCV infection and is yet to be addressed for MOPV [56]. 

We demonstrated that autophagy is implicated in the release of MOPV and LASV infectious viral particles, as silencing of the essential autophagy gene, *ATG5*, which prevents autophagosome formation, was sufficient to significantly decrease the amount of viral RNA and the infectious titer in the supernatants of both MOPV- and LASV-infected cells. In contrast, the use of CQ, which inhibits autophagosome recycling, specifically impaired MOPV infectious particle release. These results highlight a previously unknown role of autophagy in MOPV and LASV infection. Indeed, it shows that the autophagosome elongation step orchestrated by ATG5 is a positive regulator of the late phases of MOPV and LASV cycles. Interestingly, the inhibition of autophagosome maturation only decreased MOPV infectious particle production, highlighting the importance of the complete autophagy flux for MOPV infectivity, at least in HeLa cells. The viral entry assays performed during MOPV infection show that CQ did not impact the viral entry steps, therefore indicating that the observed effects of the drug in MOPV infectious particle production is linked to the inhibition of degradation steps of autophagy rather than a block of viral entry due to endosome acidification impairment. We did not perform this control with LASV, as CQ did not affect LASV infectious particle production. In summary, MOPV induces autophagy that must be productive to optimize the late steps of the viral cycle. In contrast to MOPV, LASV does not induce autophagy, but requires the initial steps of vesicle formation to also favor the late steps of the viral cycle. Given that autophagy did not appear to be blocked during LASV infection, it would be of interest to investigate whether LASV is able to escape from the last steps of autophagy degradation. We suggest that autophagy may be required for the MOPV and LASV life cycle at different steps. Many viruses have developed strategies to induce autophagy and hijack the process for their own benefit. This is true of the measles virus, HIV-1, vesicular stomatitis virus, dengue virus, hepatitis C virus, and others [56,57,58,59,60]. The late steps of the MOPV cycle, including component assembly, genome encapsidation, and budding, may occur in the acidic mature autophagosome, such as during poliovirus infection in HeLa cells [61]. LASV may take advantage of the early basal steps of autophagy to increase the processing of the matrix proteins, explaining the colocalization of LASV Z protein within autophagosomes. LASV Z protein could also subvert autophagosome membranes to improve viral egress. These phenomena have been observed during other viral infections. Indeed, autophagy has been shown to have a dual role during HIV infection. The virus first benefits from early and non-degradative basal autophagy to increase Gag processing and HIV yields in macrophages and then inhibits the degradative steps of autophagy to avoid its degradation [55]. Another virus, hepatitis B virus, induces an incomplete autophagic process in hepatoma cells that is required for HBV envelopment [62]. In addition to Mammarenavirus, the Arenaviridae family includes three other genera: hartmaniviruses and reptarenaviruses are hosted by snakes and antennaviruses are hosted by fish [63,64,65]. In contrast to mammarenaviruses and reptarenaviruses, antennavirus and hartmanivirus genomes do not contain genes encoding the Z protein, indicating that the Z interaction with the autophagy machinery described here is not a common feature among all arenaviruses.

Differences in the induction of autophagy during LASV and MOPV infection could also play a role in the immunogenicity induced by these viruses. We previously showed that MOPV strongly activates CD8^+^ T cells in an in vitro model of dendritic-cell (DC)-T-cell coculture, resulting in a strong proliferative response and the acquisition of effector and memory phenotypes, whereas the same responses were much weaker for LASV [66]. Autophagy plays a critical role in the adaptive immune response by delivering virus-derived peptides for presentation by major histocompatibility complex 1 (MHC-I) molecules to CD8^+^ T lymphocytes in a process called cross presentation [35]. As MOPV induces autophagy, this process could mediate the delivery of MOPV antigens to MHC-I at the DC surface and thus favor the subsequent activation of a specific CD8^+^ T-cell response and the killing of MOPV-infected cells. Overall, we hypothesize that, although basal autophagy is sufficient to sustain LASV infection, MOPV requires an increase in autophagy flux to replicate. Autophagy-mediated evolutionary pressure may have selected autophagy-inducing MOPV variants with higher fitness. However, such a positive effect of autophagy on MOPV replication is counterbalanced, as autophagy may also improve antigen presentation and subsequent killing of infected cells by the immune system, allowing the control of MOPV replication. In contrast, LASV is not affected by this adverse effect as the virus does not induce autophagy. This hypothesis is currently under study and could highlight the key role of autophagy in the pathogenicity of LASV and could perhaps be extended to other pathogenic arenaviruses. 

## 5. Conclusions

In summary, we used Y2H screening to identify new targets of the MOPV and LASV Z matrix proteins, identifying TAX1BP1 and NDP52 as novel interactors of the Z protein of both viruses. Functional analysis of these interactors has provided insights into the role of the autophagy machinery during the course of MOPV and LASV infection. Autophagy is transiently induced in MOPV- but not LASV-infected cells. Although the early steps of autophagy play a positive role in the production of both LASV and MOPV infectious particles, the degradative steps of autophagy are only required for MOPV infectious particle release. These results suggest that autophagy could have different roles in LASV and MOPV infection, which could explain, in part, their different pathogenicity.

## Figures and Tables

**Figure 1 viruses-11-00293-f001:**
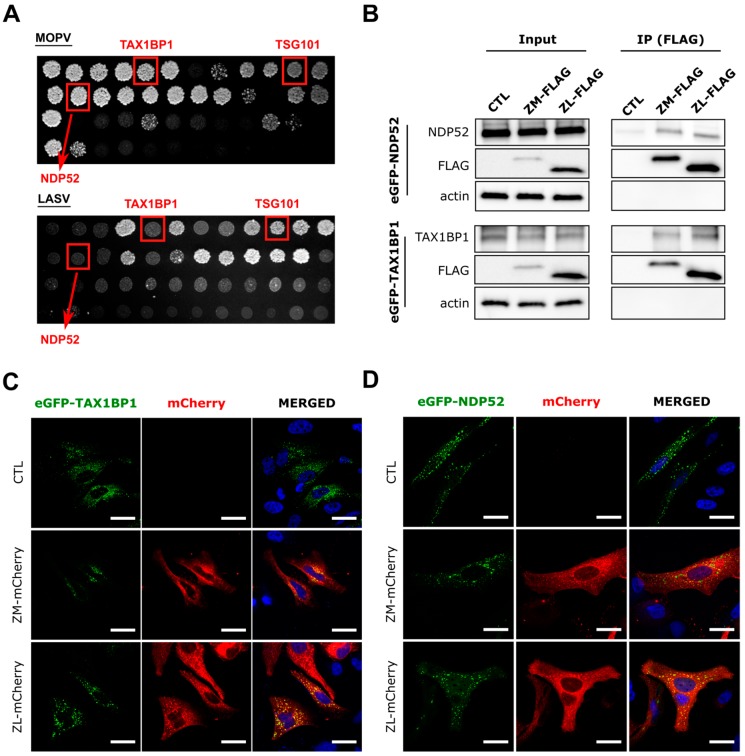
Confirmation of the interaction between MOPV and LASV Z protein and the host cell proteins NDP52 and TAX1BP1. (**A**) images of plates with “gap-repair colonies” (performed in duplicate). Clones were plated onto selective media (-L-W-H + 3AT) and left to grow for two weeks; (**B**) extracts from 293T cells cotransfected with the indicated expressing plasmids for 15 h were immunoprecipitated (IP) with FLAG magnetic beads. Exogenous eGFP-NDP52 and eGFP-TAX1BP1 were detected by Western blotting (*n* = 3 independent experiments). (**C**,**D**) HeLa cells were cotransfected with the indicated plasmids for 15 h and fixed for confocal microscopy. Exogenous eGFP-NDP52 and eGFP-TAX1BP1 are shown in green and the Z-mCherry viral proteins in red (*n* = 3 independent experiments). All images were taken on a confocal Zeiss LSM 510 with an Axioscope 63× oil immersion lens objective. Scale bar represents 30 µm.

**Figure 2 viruses-11-00293-f002:**
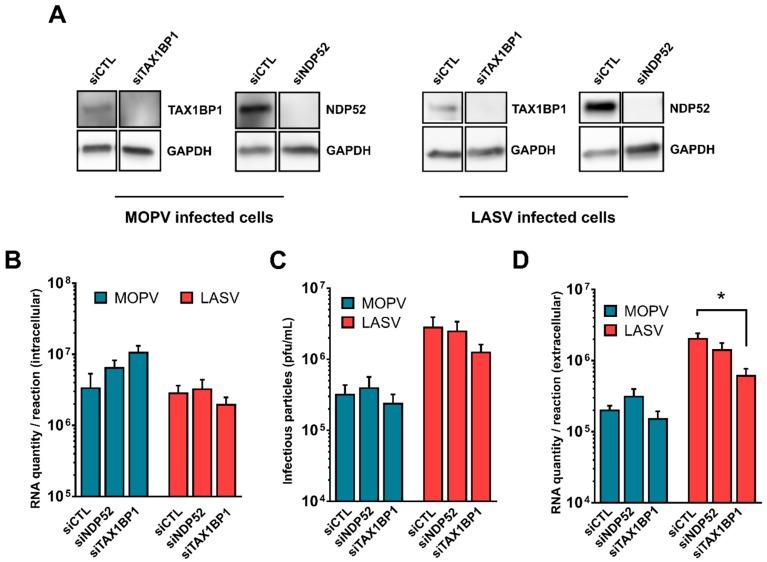
NDP52 and TAX1BP1 are neither involved in the replication nor the release of MOPV or LASV infectious particles. (**A**) HeLa cells were transfected with the indicated siRNA 72 h before analysis of silencing efficiency by Western blotting (*n* = 4 independent experiments). (**B**–**D**) the same cells as in (A) were then infected with LASV or MOPV with an MOI of 0.1 for 1 h before being maintained for three days at 37 °C. Viral RNA was then extracted from the cells and supernatants for quantification by RTqPCR. Infectious particles from the supernatants were also titrated on Vero cells. The error bars represent the standard error of the means from four independent experiments. * indicates *p* < 0.05, as determined by the Mann–Whitney test.

**Figure 3 viruses-11-00293-f003:**
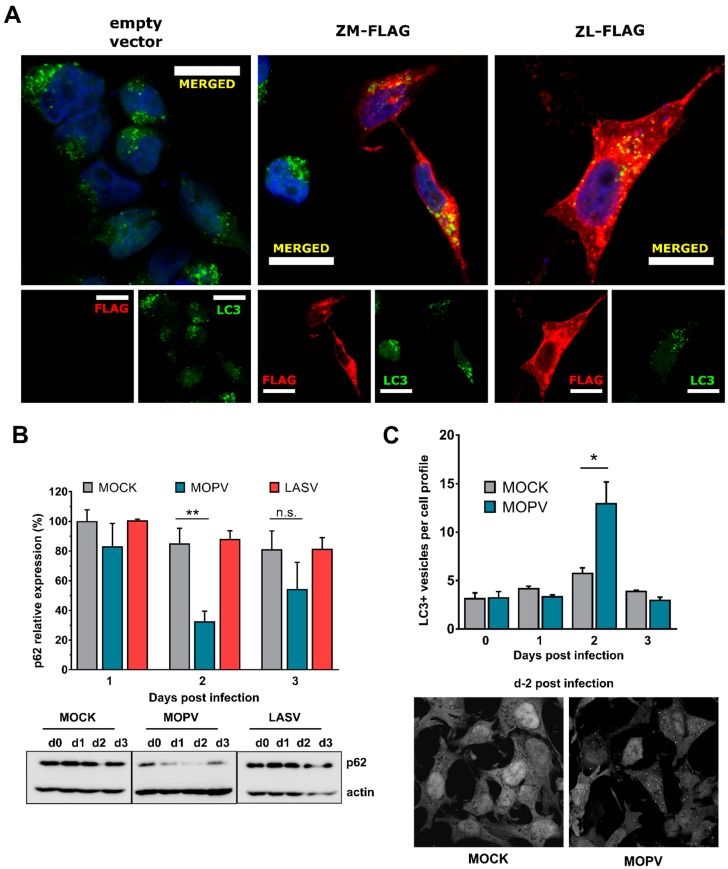
MOPV, but not LASV, induces transient autophagy in HeLa cells. (**A**) GFP-LC3 HeLa cells were treated for 2 h with CQ at a final concentration of 50 µM before transfection with the indicated plasmids for 15 h. Cells were then fixed and stained with a primary mouse anti-FLAG antibody and a secondary anti-mouse coupled Alexa555 antibody before observation by confocal microscopy (*n* = 3 independent experiments). All images were acquired using a confocal Zeiss LSM 510 microscope with an Axioscope 63× oil immersion lens objective. The scale bar represents 20 µm. (**B**) HeLa cells were mock infected or infected with LASV or MOPV at an MOI of 2. Cells were harvested at the indicated timepoints for p62 analysis by Western blotting. The graph represents the intensity of p62 over actin expression normalized to the MOCK-infected condition at d0 (not represented on the graph). The error bars represent the standard error of the means from five and four independent experiments for MOPV and LASV, respectively. **p* < 0.05, ***p* < 0.01, and n.s.: non-significant, as determined by a Student’s *t*-test. (**C**) GFP-LC3 HeLa cells were infected with MOPV at an MOI of 2 for the indicated times and fixed for confocal microscopy analysis. The images were acquired using the same microscope as in (A) with an Axioscope 63× oil immersion lens objective. The error bars represent the standard error of the means from three independent experiments, **p* < 0.05, as determined by a Student’s *t*-test. GFP-LC3 dots were counted in 50 cells per condition.

**Figure 4 viruses-11-00293-f004:**
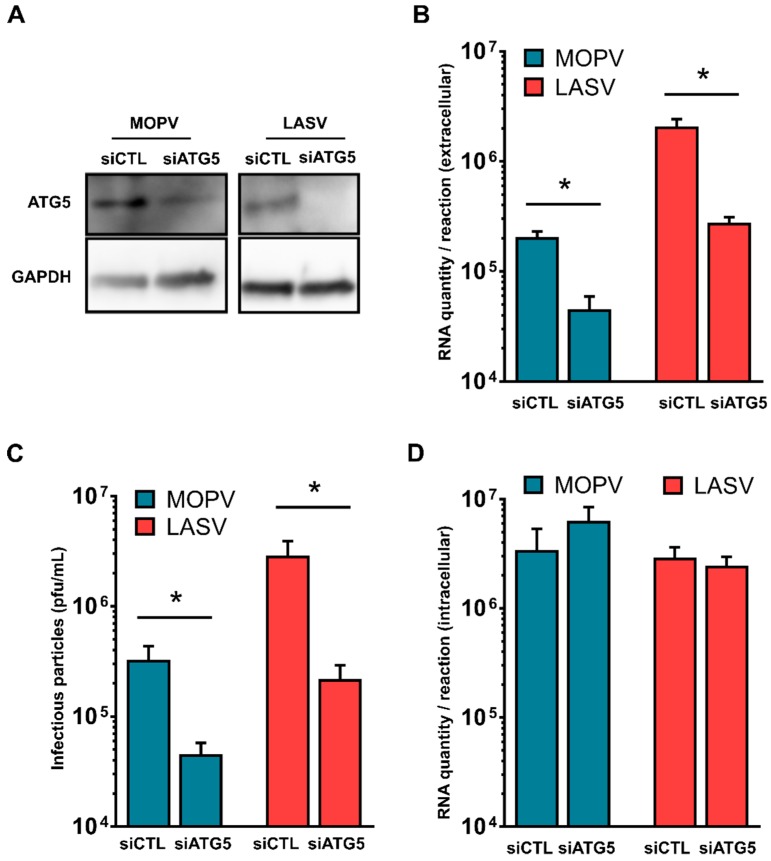
Non-degradative steps of autophagy increase both LASV and MOPV infectious particle production. (**A**) HeLa cells were transfected with the indicated siRNA for 72 h before analysis of silencing efficiency by Western blotting (*n* = 4 independent experiments); (**B**–**D**) the same cells as in (A) were then infected with LASV or MOPV at an MOI of 0.1 for 1 h before being maintained for three days at 37 °C. Viral RNA was then extracted from the cells and supernatants for quantification by RTqPCR. The infectious particles from supernatants were also titrated on Vero cells. The error bars represent the standard error of the means from four independent experiments. * indicates *p* < 0.05, as determined by the Mann–Whitney test.

**Figure 5 viruses-11-00293-f005:**
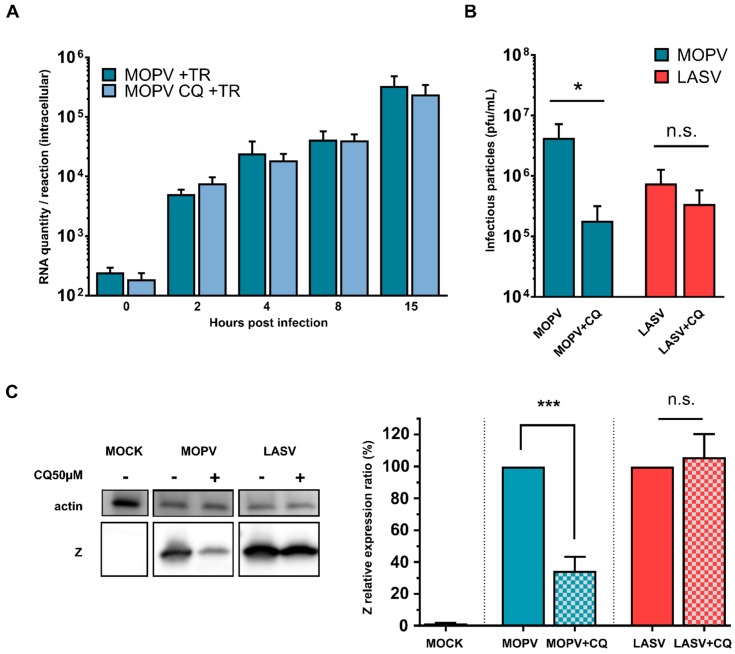
Degradative steps of autophagy increase MOPV infectious particle production. (**A**) Hela cells were pretreated with 50 µM CQ for 2 h and placed on ice before infection with MOPV at an MOI of 2. Cells were then placed on ice before proceeding to heat shock (37 °C). At the indicated timepoints, attached virus was removed by trypsin treatment (TR), and viral intracellular RNA harvested for RTqPCR analysis. (**B**) HeLa cells were infected with LASV or MOPV at an MOI of 0.1 and then treated, or not, with 50 µM CQ one day after infection for two days. Cell supernatants were then harvested and titrated on Vero cells. The error bars represent the standard error of the means from four independent experiments. * indicates *p* < 0.05, n.s.: non-significant, as determined by the Mann–Whitney test. The error bars represent the standard error of the means from four independent experiments. (**C**) The same cells as in (B) were lysed and the quantity of Z protein inside the cells has been measured by Western blotting. Representative results are shown, along with a graph representing the intensity of the Z protein bands over actin. The error bars represent the standard error of the means from four independent experiments. n.s. non-significant; *** *p* < 0.001, as determined by Student’s *t*-test.

**Table 1 viruses-11-00293-t001:** Selected “autophagy-linked” host protein interactors of LASV and MOPV Z identified by Y2H and validated by gap-repair. The first and second columns correspond to the canonical gene names and gene IDs, respectively, for the selected interacting cellular proteins. Columns 3 and 4 provide a summarized role of the corresponding protein in autophagy pathways and the reference, respectively. Column 5 shows the number of positive yeast colonies obtained for each cellular protein for the indicated virus Z protein (retrieved from mouse brain, human spleen cDNA libraries and ORFeome). Column 6 shows the results obtained for the validation step by gap-repair in yeast. The “x” indicates the presence of growing colonies for the selected condition. Tsg101 has already been described by others and served as a positive control.

Gene	Gene ID	Role in Autophagy	Ref.	HT-Y2H Screening Hits	Screening Validation by Gap-Repair
Z MOPV	Z LASV	Z MOPV	Z LASV
TSG101	7251	Y2H Positive control	[19]	13	36	x	x
TAX1BP1	8887	Selective autophagy adaptors involved in selective degradation of pathogens by autophagy, or xenophagy	[44]	60	0	x	
NDP52	10241	[45]	65	0	x

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
