# Peer review of "Autophagy Promotes Infectious Particle Production of Mopeia and Lassa Viruses"

_viruses, 2019, doi:10.3390/v11030293_

Round 1
Reviewer 1 Report
Viruses - MDPI
Manuscript ID: 459226
Screening of Lassa and Mopeia virus Z matrix protein interactors reveals a positive role for autophagy machinery in the production of infectious particles
By Nicolas Baillet et al.
This is a well-written manuscript seeking to assess role of autophagy in production of infection particles during replication of two mammalian arenaviruses, Lassa (LASV) and Mopeia (MOPV). These viruses are hosted by the same natural host species, M. natalensis, genetically closely related, and can produce reassortants after co-infection. However, LASV is a human pathogen and can cause fatal Lassa fever disease, while MOPV is non-pathogenic and can protect non-human primates against LASV challenge. Dr. Baize’s group made substantial contribution in the field by comparing host responses in vitro and in vivo during LASV vs. MOPV infections. The manuscript is a continuation of their efforts in this area. Using yeast-two-hybrid screening, co-immunorecipitation and co-localization assays, the authors identified two autophagy adaptors, NDP52 and TAX1BP1, involved in interaction with Z protein of LASV and MOPV. Reduced expression of these adaptors did not affect replication of the infectious MOPV and LASV indicating that these factors are not required for virus replication. Still, the authors showed that autophagy was transiently induced by MOPV, but not LASV, and suggested that this difference could play a role in different immune responses induced by these viruses. Several issues have to be addressed before accepting the manuscript for publication.
1. Title. Since NDP52 and TAX1BP1 silencing with appropriate siRNAs did not affect LASV and MOPV replication in HeLa cells (Fig. 2B), the authors correctly concluded that these factors are not required for replication of these viruses. However, the title claims “a positive role for autophagy machinery in the production of infectious particles”. This title is biased and has to be corrected in line with provided results.
2. Introduction. The authors have to use updated terminology according to recent changes in taxonomic status of LASV and MOPV. These viruses belong to mammalian arenaviruses. In addition to Mammarenavirus, the Arenaviridae family contains 3 genera to host viruses infect fish (antennaviruses) and snakes (hartmaniviruses and reptarenaviruses) (Radoshitzky et al., 2015; Maes P et al., 2018, 2019). Notably, and it has to be important for this manuscript, antennaviruses and hartmaniviruses have no genes encoding Z protein which is encoded by mammarenaviruses and reptarenaviruses. It indicates that Z interaction with autophagy machinery described in the manuscript is not common features of arenaviruses. It has to be more elaborated in Discussion section.
3. Introduction, line 36-38. The Ref. 1 is incorrectly referenced and interpreted. The authors stated “The disease is a significant cause of morbidity (tens of thousands cases annually) and mortality (hundreds of fatalities annually) and is constrained to sub-Saharan West Africa”. In fact, Joe McCormick and co-workers in “A prospective Study of the Epidemiology and Ecology of Lassa Fever” provided the incidence of seroconversion (5-22%/yr) and illness/infection ratio (9-26%). They concluded that “if our data are representative of the situation in West Africa, as many as 100,000-300,000 infections may occur each year, with 5,000 deaths”. These numbers, are very outdated and probably are underestimated (based only on LASV/Josiah-based sero-diagnostics). Still, these numbers are widely accepted in the field by experts and by international organizations (e.g., WHO, CEPI and etc.).
4. Line 45, “Mopeia virus (MOPV) also belongs to the Old-World arenavirus clade”. It is wrong statement. There is no such clade. In fact, MOPV belongs to the OW sero-group of mammalian arenaviruses. It is very closely related to LASV not only genetically (can make replication-competent reassortants) but also ecologically since it has hosted by the same rodents, M. natalensis.
5. Lane 138, “… cells were reverse transfected…” It is jargon, please use appropriate terminology.
6. Lane 294, “…MOPV/LASV infection.” It is confusing abbreviation, can be wrongly consider as an infection with MOPV/LASV reassortant. Has to be corrected.
7. The authors correctly mentioned limitation of their Y2H screening system. There are potentially other host factors linked to autophagy and involved in LASV and MOPV replication and/or virus egress. It will be helpful to include in the Discussion recent publication describing role of BAG3 host protein in LASV VLP egress (Han Z et al., 2018). BAG3 is a stress-induced molecular co-chaperone that functions to regulate cellular protein homeostasis and cell survival via Chaperone-Assisted Selective Autophagy.
Author Response
Dear reviewers,
As you will see, we have also submit a version in which the changes are highlighted in order that you will be able to easily track them. In addition to the modifications requested by the reviewers, we took advantage of the revision to improve the English style of the manuscript. Of course, this improvement do not change the content and the meanings of the paper.
Review 1
1. Title. Since NDP52 and TAX1BP1 silencing with appropriate siRNAs did not affect LASV and MOPV replication in HeLa cells (Fig. 2B), the authors correctly concluded that these factors are not required for replication of these viruses. However, the title claims “a positive role for autophagy machinery in the production of infectious particles”. This title is biased and has to be corrected in line with provided results.
a. We agree with the remark and have replaced the previous title by the following: “Autophagy promotes infectious particle production of Mopeia and Lassa viruses.”
2. Introduction. The authors have to use updated terminology according to recent changes in taxonomic status of LASV and MOPV. These viruses belong to mammalian arenaviruses. In addition to Mammarenavirus, the Arenaviridae family contains 3 genera to host viruses infect fish (antennaviruses) and snakes (hartmaniviruses and reptarenaviruses) (Radoshitzky et al., 2015; Maes P et al., 2018, 2019). Notably, and it has to be important for this manuscript, antennaviruses and hartmaniviruses have no genes encoding Z protein which is encoded by mammarenaviruses and reptarenaviruses. It indicates that Z interaction with autophagy machinery described in the manuscript is not common features of arenaviruses. It has to be more elaborated in Discussion section.
a. Thank you for the comment. The terminology has been updated in the Introduction section. The second part of your comment and the description of the different genera of arenaviruses have been added to our Discussion section in order to correctly precise that our findings are not a common feature of all Arenaviridae family. Here the modification (line 491) : “In addition to Mammarenavirus, the Arenaviridae family contains three other genuses, including hartmanivirus and reptarenavirus (hosted by snakes) as well as recently discovered antennaviruses (hosted by fishes) [63–65]. In contrast with mammarenaviruses and reptarenaviruses, antennaviruses and hartmaniviruses have no genes encoding the Z protein, indicating that Z interaction with autophagy machinery described here is not a common feature of arenaviruses”.
3. Introduction, line 36-38. The Ref. 1 is incorrectly referenced and interpreted. The authors stated “The disease is a significant cause of morbidity (tens of thousands cases annually) and mortality (hundreds of fatalities annually) and is constrained to sub- Saharan West Africa”. In fact, Joe McCormick and co-workers in “A prospective Study of the Epidemiology and Ecology of Lassa Fever” provided the incidence of seroconversion (5-22%/yr) and illness/infection ratio (9-26%). They concluded that “if our data are representative of the situation in West Africa, as many as 100,000-300,000 infections may occur each year, with 5,000 deaths”. These numbers, are very outdated and probably are underestimated (based only on LASV/Josiah-based sero-diagnostics). Still, these numbers are widely accepted in the field by experts and by international organizations (e.g., WHO, CEPI and etc.).
a. The precise estimation of the number of cases and deaths have been replaced in the appropriate section, in order to comply with the reference (line 36).
4. Line 45, “Mopeia virus (MOPV) also belongs to the Old-World arenavirus clade”. It is wrong statement. There is no such clade. In fact, MOPV belongs to the OW sero-group ofammalian arenaviruses. It is very closely related to LASV not only genetically (canake replication-competent reassortants) but also ecologically since it has hosted by theame rodents, M. natalensis.
a. This sentence has been replaced by the following (line 44):” Mopeia virus (MOPV) also belongs to the Old-World complex of mammalian arenaviruses and is very closely related to LASV in addition to be hosted by the same rodents, Mastomys natalensis.”
5. Lane 138, “… cells were reverse transfected…” It is jargon, please use appropriate terminology.
a. This sentence has been replaced by the following (line 141): “Freshly passaged HeLa cells were added to pre-plated transfection media containing the indicated siRNA, at a final concentration of 20 nM and using Lipofectamine RNAiMAX (Invitrogen), according to the manufacturer’s instructions.”
6. Lane 294, “…MOPV/LASV infection.” It is confusing abbreviation, can be wrongly consider as an infection with MOPV/LASV reassortant. Has to be corrected.
a. This sentence has been replaced by “MOPV or LASV infection”.
7. The authors correctly mentioned limitation of their Y2H screening system. There are potentially other host factors linked to autophagy and involved in LASV and MOPV replication and/or virus egress. It will be helpful to include in the Discussion recent publication describing role of BAG3 host protein in LASV VLP egress (Han Z et al., 2018). BAG3 is a stress-induced molecular co-chaperone that functions to regulate cellular protein homeostasis and cell survival via Chaperone-Assisted Selective autophagy.
a. Thank you for this suggestion. This reference is now discussed line 412 “Other host factors interacting with LASV Z protein and possibly linked to autophagy have been unraveled by others [48]. In this study, authors have identified the WW-domain bearing protein BCL2 Associated Athanogene 3 (BAG3) as a negative regulator of Lassa VLP egress. BAG3 has several functions in cell cycle including a stress-induced role in chaperone-assisted selective autophagy (CASA). Although this study appears contradictory with the proviral role of autophagy described here during LASV infection, it reinforces the link existing between Mammarenavirus infection and autophagy pathways.”
Reviewer 2 Report
Since this manuscript is focused on the Z protein, please include the reference below in line 51. It is the first description of the Z gene. Salvato MS, Shimomaye EM. The completed sequence of lymphocytic choriomeningitis virus reveals a unique RNA structure and a gene for a zinc finger protein. Virology. 1989 Nov;173(1):1-10.
Reference 8 in line 47 is not appropriate. This reference refers to a modified version of MOPV called MOPEVACLASV. Please clarify this on the text or remove reference 8.
Line 89. Please add (Y2H) after Yeast two-hybrid.
Line 171. Please explain Why viral entry assays were performed only for MOPV?
Figure 1A. This figure is very confusing. Please identify columns and rows, specifying what is in each of them.
Line 295. Why did the researchers use such a high MOI? Considering interference particles, did you contemplate the possibility of having false results due to diffefences in the presence of defective particles?
In the experiment shown in figure 3. Why you did not include LASV?
Figure 4A and line 328, the authors stated that there was an efficient silencing of ATG5. However, there is still a fainted band in MOPV in contrast to LASV. Is it because the silencing was incomplete? Or MOPV induces ATG5 expression? Could this difference affect the results and conclusion?
Figure 5A. Why you did not include LASV?
Line 487-89 the authors discuss the pathogenicity in humans but what happened with the autophagy pathway in the reservoir in which both viruses establish chronic infections?
Reference 30 in line 585 is incomplete. Please add missing information (Journal, year…)
Author Response
Dear reviewers,
As you will see, we have also submit a version in which the changes are highlighted in order that you will be able to easily track them. In addition to the modifications requested by the reviewers, we took advantage of the revision to improve the English style of the manuscript. Of course, this improvement do not change the content and the meanings of the paper.
Review 2
1. Since this manuscript is focused on the Z protein, please include the reference below in line 51. It is the first description of the Z gene. Salvato MS, Shimomaye EM. The completed sequence of lymphocytic choriomeningitis virus reveals a unique RNA structure and a gene for a zinc finger protein. Virology. 1989 Nov;173(1):1-10.
a. You’re right. The reference has been added to the appropriate line.
2. Reference 8 in line 47 is not appropriate. This reference refers to a modified version of MOPV called MOPEVACLASV. Please clarify this on the text or remove reference 8.
a. The sentence has been clarified by the following one (line 47): ”MOPV has even been shown to confer protection against challenge with LASV in non-human primates and therefore represents a useful platform to generate protective vaccines against Lassa fever.” One reference (Lukashevich et al. 2005) has also been added.
3. Line 89. Please add (Y2H) after Yeast two-hybrid.
a. The correction has been made.
4. Line 171. Please explain Why viral entry assays were performed only for MOPV?
a. We did not performed viral entry assay on LASV infection because this assay was done as a control to confirm that CQ effect on MOPV replication was due to impairment of autophagy rather than blocking of endosomal acidification, thus viral entry. As LASV do not induce autophagy in cells and because CQ did not affect LASV infectious particle production, there was no need to perform such a control with LASV. Please notice that sentences have been added to the discussion section (line 468) to clarify this point.
5. Figure 1A. This figure is very confusing. Please identify columns and rows, specifying what is in each of them.
a. Unfortunately, this Gap-repair figure presents positive colonies in which other host cell proteins have interacted with MOPV and/or LASV Z protein in yeasts. These interactors are currently investigated in our laboratory and an article is in preparation. We therefore cannot mention the name of these interactors in this article. Please note that a mistake has been deleted in this figure in the new version of the manuscript. It concerns the presence of Optineurin that had been mentioned by mistake. To avoid incomprehension of the results, the indication of the protein has been deleted.
6. Line 295. Why did the researchers use such a high MOI? Considering interference particles, did you contemplate the possibility of having false results due to differences in the presence of defective particles?
a. In these experiments, we searched to investigate the induction of autophagy upon LASV or MOPV infection. To do that, it is necessary to infect a large quantity of cells in order to see the effect in most of the cells. By using a MOI of 2 for each virus, all cells are rapidly infected, allowing us to follow autophagy induction with a better synchronisation. We used a low MOI for viral production or to see the impact of an effect (silencing, drug..) on viral fitness in order to limit defective interfering particles. Please notice that we add some explanations in the discussion section (line 446).
b. As you correctly mentioned it, it is possible that our infection with a high MOI increased the number of defective particles. However, as we used the same MOI for each virus (which are closely related in terms of replication efficiency, it means ratio between viral RNA and infectious particles), our conclusion that only MOPV is able to induce autophagy seems due to the virus we used and not to the MOI and the possible effects of defective particles.
7. In the experiment shown in figure 3. Why you did not include LASV?
a. As you correctly mentioned, in the figure 3C, we counted GFP-LC3+ vesicles only in MOPV-infected GFP-LC3 HeLa cells. We unfortunately did not include LASV because of technical limits imposed by the BSL-4 containment. Indeed, in order to clearly observed and quantify autophagosomes in cells, the use of confocal microscopy is necessary. We have access to confocal microscopy when we used MOPV (under BSL-2 conditions). Unfortunately, the BSL-4 in Lyon does not contain a confocal microscope and it is impossible to take out the LASV samples from the BSL4 without keeping them two weeks into formaldehyde solution. We have determined that such a treatment importantly alters the slides, making the observation of the fluorescence signal impossible after these two weeks of fixation. A sentence explaining that has been added line 449.
8. Figure 4A and line 328, the authors stated that there was an efficient silencing of ATG5. However, there is still a fainted band in MOPV in contrast to LASV. Is it because the silencing was incomplete? Or MOPV induces ATG5 expression? Could this difference affect the results and conclusion?
a. The figure 4A is one example of four independent experiments. Please be careful, the Western blots have been made three days after ATG5 silencing in cells and before MOPV (or LASV) infection. As a consequence, it does not mean that MOPV induces ATG5 expression because cells were not infected at this time. This figure show that ATG5 was silenced in cells the day of infection by LASV or MOPV as compared to control condition. Furthermore, this phenomenon was not observed in the others experiments we made, therefore not affecting the conclusions we made.
9. Figure 5A. Why you did not include LASV?
a. Please see response 4.a.
10. Line 487-89 the authors discuss the pathogenicity in humans but what happened with the autophagy pathway in the reservoir in which both viruses establish chronic infections?
a. There is no study in which the autophagy flux and its role in the persistence of LASV in mastomys have been investigated. It would be very interesting to compare the role of autophagy during LASV infection between human cell lines and in the reservoir. Especially, it is of interest to investigate whether autophagy flux (or its absence) in mastomys could play a role in the persistence of LASV and the asymptomatic infection that is characteristic of the reservoir.
11. Reference 30 in line 585 is incomplete. Please add missing information (Journal, year…)
a. The problem in the reference has been fixed. Thank you.